**Subject Area:**
cellular biology/developmental biology/
genetics

*Drosophila*, cell migration, regeneration,
radiation, salivary gland, wing disc

**Authors for correspondence:**
Monn Monn Myat
e-mail: myat@cshl.edu
Tin Tin Su
e-mail: tin.su@colorado.edu

†Present address: Cold Spring Harbor
Laboratory, 1 Bungtown Road, Cold Spring
Harbor, NY 11724, USA.
‡Present address: Department of Cell Biology,
Emory University, 615 Michael Street, Atlanta,
GA 30322, USA.

Electronic supplementary material is available
online at https://dx.doi.org/10.6084/m9.
figshare.c.4472936.

# Regulators of cell movement during development and regeneration in *Drosophila*

Monn Monn Myat[1,†], Dheveline Louis[1], Andreas Mavrommatis[1],
Latoya Collins[1], Jamal Mattis[1], Michelle Ledru[2], Shilpi Verghese[2,‡]
and Tin Tin Su[2,3]

[1]Department of Biology, Medgar Evers College, City University of New York, Brooklyn, NY 11225, USA
[2]Department of Molecular, Cellular and Developmental Biology, University of Colorado, Boulder,
CO 80309-0347, USA
[3]University of Colorado Comprehensive Cancer Center, Anschutz Medical Campus, 13001 East 17th Place,
Aurora, CO 80045, USA

ⓘ TTS, 0000-0003-0139-4390

Cell migration is a fundamental cell biological process essential both for normal development and for tissue regeneration after damage. Cells can migrate individually or as a collective. To better understand the genetic requirements for collective migration, we expressed RNA interference (RNAi) against 30 genes in the *Drosophila* embryonic salivary gland cells that are known to migrate collectively. The genes were selected based on their effect on cell and membrane morphology, cytoskeleton and cell adhesion in cell culture-based screens or in *Drosophila* tissues other than salivary glands. Of these, eight disrupted salivary gland migration, targeting: Rac2, Rab35 and Rab40 GTPases, MAP kinase-activated kinase-2 (MAPk-AK2), RdgA diacylglycerol kinase, Cdk9, the PDSW subunit of NADH dehydrogenase (ND-PDSW) and actin regulator Enabled (Ena). The same RNAi lines were used to determine their effect during regeneration of X-ray-damaged larval wing discs. Cells translocate during this process, but it remained unknown whether they do so by directed cell divisions, by cell migration or both. We found that RNAi targeting Rac2, MAPk-AK2 and RdgA disrupted cell translocation during wing disc regeneration, but RNAi against Ena and ND-PDSW had little effect. We conclude that, in *Drosophila*, cell movements in development and regeneration have common as well as distinct genetic requirements.

## 1. Introduction

Collective cell migration is important for forming the complex architecture of tissues and organs during embryogenesis and also plays an important role in cancer progression. Examples include neural crest cells that migrate as a group and sheets of epithelial cells that migrate collectively during gastrulation [1]. Unlike single migrating cells, collectively migrating cells face the additional challenge of having to coordinate the activities of all cells in the group to achieve directed movement. Studies in vertebrate and invertebrate model organisms have identified a number of molecular features of collective cell migration. These include the identification of leader and follower cells, the molecular basis for adhesive contacts among cells and between cells and the surrounding environment, and the collective response to guidance cues. How these are regulated to allow sufficient plasticity for collective migration while maintaining tissue integrity remains an active area of research.

The *Drosophila* embryonic salivary gland is a well-established experimental system for studying collective cell migration [2–4]. The gland consists of a pair of elongated secretory tubes that are connected to the larval mouth by fine duct

tubes. Salivary gland development begins with invagination of primordial cells from the embryo surface followed by collective migration of the gland as an intact organ. There is no cell death or cell proliferation throughout gland development and the gland cells retain their epithelial characteristics during morphogenesis. Collective migration of the salivary gland occurs through coordinated migration of the distal and proximal gland cells. While distal gland cells elongate and extend actin-rich basal membrane protrusions in a process dependent on Rac GTPases, proximal gland cells change shape from columnar to cuboidal and rearrange in a Rho1 GTPase-dependent manner [5–8].

Rac/Rho-dependent intracellular changes are governed by the activity of integrin receptors at the sites of contact between salivary gland cells and between these cells and the substratum. Stable microtubules and the KASH-domain containing protein Klarsicht are responsible at least partially for localizing integrins at the contact sites [9]. Integrin adhesion receptors, $\alpha$PS1$\beta$PS (expressed in the salivary gland) and $\alpha$PS2$\beta$PS (expressed in the surrounding mesoderm) concentrate at sites of contact between the migrating gland cells and the surrounding mesoderm-derived tissues [10]. Loss of integrin expression or alteration of integrin localization results in gland migration defects with gland cells being unable to initiate or complete posterior migration, respectively [2,9]. One mechanism by which $\alpha$PS1$\beta$PS integrin controls salivary gland migration is by downregulating E-cadherin and promoting basal membrane protrusions through Rac1 in the distal gland cells [6]. Dynamin GTPase, which mediates endocytosis, is also required for E-cadherin downregulation in migrating gland cells [5], suggesting that Rac1 may act through endocytosis to downregulate E-cadherin. E-cadherin downregulation is likely to be temporally and spatially regulated in the migrating salivary gland such that cells acquire sufficient plasticity for collective migration while maintaining enough cell–cell adhesion to ensure tissue integrity.

Cell movements occur not only during normal development but also during tissue regeneration. Compared with normal development, cell movements during regeneration are even less well understood. *Drosophila* larval wing discs have been established as a system to study regeneration after a variety of damage including surgical ablation, genetic ablation through the expression of apoptotic genes, and by ionizing radiation (IR) [11]. We reported before that cells change position during regeneration of larval wing discs damaged by IR. IR induces apoptosis in the single layer epithelium of the larval wing disc. IR-induced apoptosis is scattered but not random and occurs instead in an invariant pattern [12]. We found previously that cells of the future hinge are protected from IR-induced apoptosis [12]. Some hinge cells then lose their hinge fate and translocate to the wing pouch area that suffers more IR-induced apoptosis, where the hinge cells convert to the pouch fate and participate in regeneration of the pouch. Signalling through *Drosophila* STAT92E (homologue of STAT3/5) and Wingless (homologue of Wnt1) are required cell autonomously for IR-induced regenerative behaviour; knocking down each with RNAi or genetic inhibitors (e.g. Axin against Wg) prevented the translocation and fate change by the hinge cells [12]. Using this model, we have uncovered the requirement for epigenetic regulators of IR-induced fate change and translocation [13]. But cell biological mechanisms by which former hinge cells translocate from the hinge into the pouch remained completely unknown. We do not know even whether the hinge cells migrate as opposed to use directed cell divisions that 'push' daughter cells towards the pouch.

To better understand how cellular adhesion and the cytoskeleton control collective cell migration, we used RNA interference (RNAi) against a focused group of 30 genes known or predicted to affect cell or membrane morphology, adhesion and cytoskeleton. We identified eight lines that, when expressed specifically in the salivary gland, disrupt gland migration. These include four genes with previously unknown roles in collective migration of the salivary gland. To address whether common genetic requirements contribute to cell migration in salivary glands and cell position changes during regeneration in the wing disc, we tested a subset of the RNAi lines also in the wing disc. The results identified genes with previously unknown roles in regeneration and suggest that epithelial cell movements during development and regeneration have overlapping as well as distinct genetic requirements.

## 2. Results

To test for a cell-autonomous requirement in the salivary glands, we generated a recombinant *Drosophila* line with UAS-GFP$^{NLS}$ (green fluorescent protein (GFP) tagged with 14 nuclear localization signals) and the *fkh*-GAL4 driver (for gland-specific expression) on the third chromosome. This recombinant line was then crossed with each of 30 RNAi lines (30 genes). We chose the genes based on previous reports that depletion by RNAi of these genes led to defects in cell or membrane morphology, cytoskeleton or cell adhesion in *Drosophila* cell culture or in tissues other than the salivary gland [14–17] (electronic supplementary material, table S1). Another selection criterion was for genes for which transgenic lines were available at the time the project was initiated and had RNAi constructs inserted on the second chromosome. This was in case we needed to generate a stock that expressed both RNAi (chromosome II) and UAS-GFP$_{NLS}$, *fkh*-GAL4 (chromosome III) in the future. For each RNAi line tested, we analysed the shape of GFP-expressing glands in a minimum of 20 live embryos. If at least five GFP-expressing glands showed abnormal morphology indicative of a possible migration defect, then the phenotype was confirmed in fixed embryos immunostained for the salivary gland marker protein dCREB-A. Approximately 20 stage 14 glands each were analysed in fixed embryos and the penetrance ranged from 5 to 10 defective glands per sample.

Of the 30 RNAi lines, we identified eight that inhibited gland migration when expressed cell autonomously, targeting: *enabled*, which encodes an actin-binding protein; *rdgA*, which encodes a diacylglycerol kinase; MAP kinase-activated kinase 2 (MAPk-AK2); *ND-PDSW*, which encodes a subunit of NADH dehydrogenase (to be called PDSW hereafter); cyclin-dependent kinase 9 (Cdk9); and *Rac2, Rab35* and *Rab40*, which encode small GTPases. Rac and Rab GTPases are already known to be required for salivary gland migration (for example, [5,6]). Cdk9 has a well-documented role in general transcription [18] so defects due to its depletion may be indirect. Therefore, we focused our analysis on PDSW, RdgA, Enabled (Ena) and MAPk-AK2 to understand their roles in collective migration of the salivary gland.

royalsocietypublishing.org/journal/rsob    Open Biol. 9: 180245

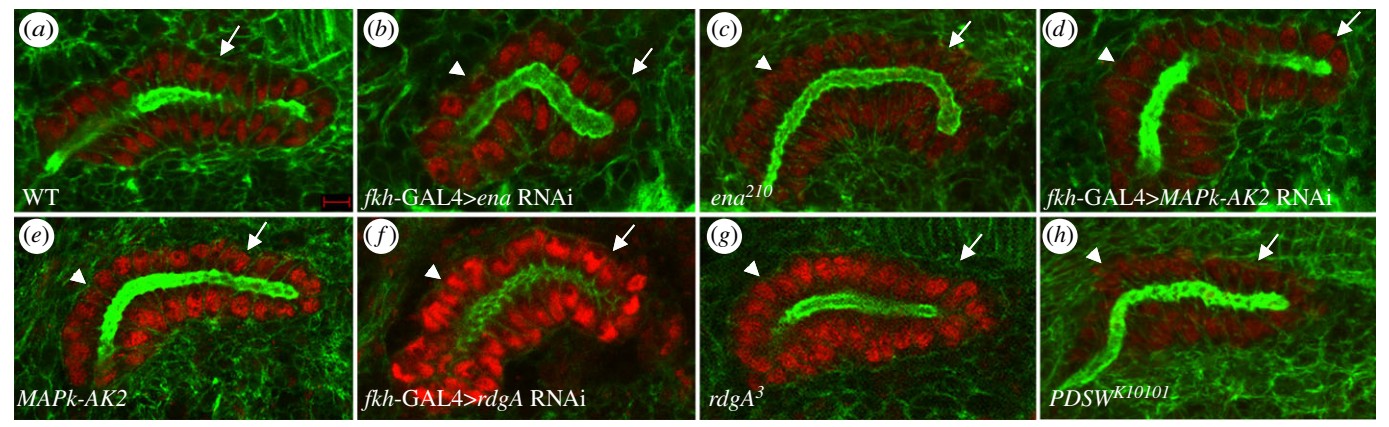

**Figure 1.** Salivary gland migration and rearrangement defects after depletion of Ena, MAPk-AK2, RdgA and PDSW. (*a*) In wild-type embryos at stage 14, the salivary gland has turned and completed its posterior migration (arrow). (*b–h*) In glands expressing *ena* RNAi (*b*), embryos homozygous for *ena²¹⁰* (*c*), glands expressing *MAPk-AK2* RNAi (*d*), embryos homozygous for *MAPk-AK2^G265* (*e*), glands expressing *rdgA* RNAi (*f*), embryos homozygous for *rdgA³* (*g*) or embryos homozygous for *PDSW^k10101* (*h*), the distal gland has turned posteriorly (arrows), but the proximal gland failed to turn and is still oriented dorsally or 'up' in the images (arrowheads). All embryos shown are at stage 14 and were labelled for F-actin (green) and dCREB (red) except for the embryo in (*f*), which was labelled for DE-cadherin (green) and dCREB (red). Scale bar in (*a*) represents 5 μm. Images are shown dorsal up and posterior right.

## 2.1. Salivary glands failed to complete posterior migration in *enabled, rdgA, MAPk-AK2* and *PDSW* mutant embryos

In wild-type embryos, cells of the salivary gland form a tube that first elongates dorsally before starting to migrate posteriorly. Posterior migration begins with cells at the distal end of the salivary gland but by embryonic stage 14 both the distal and proximal parts of the salivary glands have turned and migrated posteriorly (figure 1*a*; [19]). By contrast, in embryos of the same stage where dsRNA against *ena* was expressed specifically in the salivary gland, the distal gland turned and migrated posteriorly whereas the proximal gland did not (figure 1*b*). To confirm the gland migration defect observed with RNAi knockdown, we analysed *ena* zygotic loss of function mutant embryos for gland migration defects.

*ena²¹⁰* is a single C to T change resulting in an A97 V change in the conserved EVH1 (enabled/VASP homology 1) domain; *ena²³* is an A to G change resulting in an N379F change in the proline-rich domain as well as an A to T change resulting in a stop codon to delete the EVH2 domain; *ena^02029* is a transposition insertion in the 5′-UTR [20]. In embryos homozygous mutant for *ena²¹⁰* or *ena²³*, salivary glands failed to migrate completely, with the distal gland turned posteriorly but not the proximal gland (figure 1*c* and data not shown). This phenotype is similar to glands where *ena* was knocked down with RNAi. During posterior migration, proximal gland cells rearrange to form a narrower and more elongated tube [7]. Proximal glands of *ena²³* and *ena^02029* mutant embryos showed an average of 10.0 ± 0.8 nuclei per cross section ($n = 7$ glands) compared with 8.0 ± 1.1 in wild-type ($n = 3$ glands; for examples of nuclear density changes, see fig. 2 of [21] and electronic supplementary material, figure S1). The difference was significant ($p = 0.009$), suggesting that proximal gland cells failed to rearrange in *ena* mutant embryos.

We also analysed salivary gland migration defects in embryos homozygous for *MAPk-AK2, PDSW* and *rdgA* mutations and wild-type embryos where *MAPk-AK2* or *rdgA* have been knocked down specifically in the gland. *PDSW^k10101* was generated by P-element insertional mutagenesis (Flybase), and *rdgA³* by EMS mutagenesis [22]. Similar to knockdown and loss of function of *ena*, depletion of *MAPk-AK2, PDSW* and *rdgA* resulted in gland migration defects where the distal gland turned posteriorly but the proximal gland did not (figure 1*d–h*). However, unlike in *ena* mutants, quantification of the number of nuclei in *PDSW* mutant embryos and glands with *rdgA* knocked down showed no statistically significant differences (data not shown).

## 2.2. Ena localization and salivary gland lumen defects

To better understand how *ena* contributes to cell migration during salivary gland development, we determined the subcellular localization of Ena by immunostaining migrating salivary glands of wild-type embryos. At embryonic stage 12 when the gland is beginning to turn posteriorly, Ena was found in discrete foci sub-apical to the apical domain marker protein, DaPKC (figure 2*a*). At stage 14, when the gland had turned posteriorly, Ena localized in a zonular manner sub-apical to DaPKC (figure 2*b*). Ena also localized to the basal domain of migrating gland cells (figure 2*b,b′*). Because Ena localized to the apical domain, we determined if the loss of *ena* affected the lumen morphology and/or the localization of apical proteins. Analysis of the gland lumen with F-actin labelling showed that the lumen of *ena²¹⁰* mutant embryos was irregularly shaped in the distal and medial parts of the gland compared with heterozygous siblings (figure 3*a–c*). Immunostaining for the apical protein DaPKC (figure 3*d,e*) and the apical–lateral protein DE-cadherin (DE-cad; figure 4*a,b*) showed that loss of *ena* did not affect the apical localization of these proteins. Measurements of the apical domain area showed no difference between *ena* mutant gland cells and wild-type cells (data not shown). However, apical domains of *ena* mutant gland cells did not elongate in the proximal–distal direction to the same extent as in wild-type gland cells (figure 4*d*, quantified from confocal images). Similar to *ena* mutant gland cells, the apical domain area in *PDSW* mutant gland cells was not different from that in wild-type cells (data not shown); however, the apical domain failed to elongate in the proximal–distal direction (figure 4*c,d*) and irregularities in the lumen shape at the distal tip were also observed (figure 4*c*). RNAi against *MAPk-AK2* or *rdgA* did not produce apparent lumen defects; therefore, the apical domain was not analysed in these embryos.

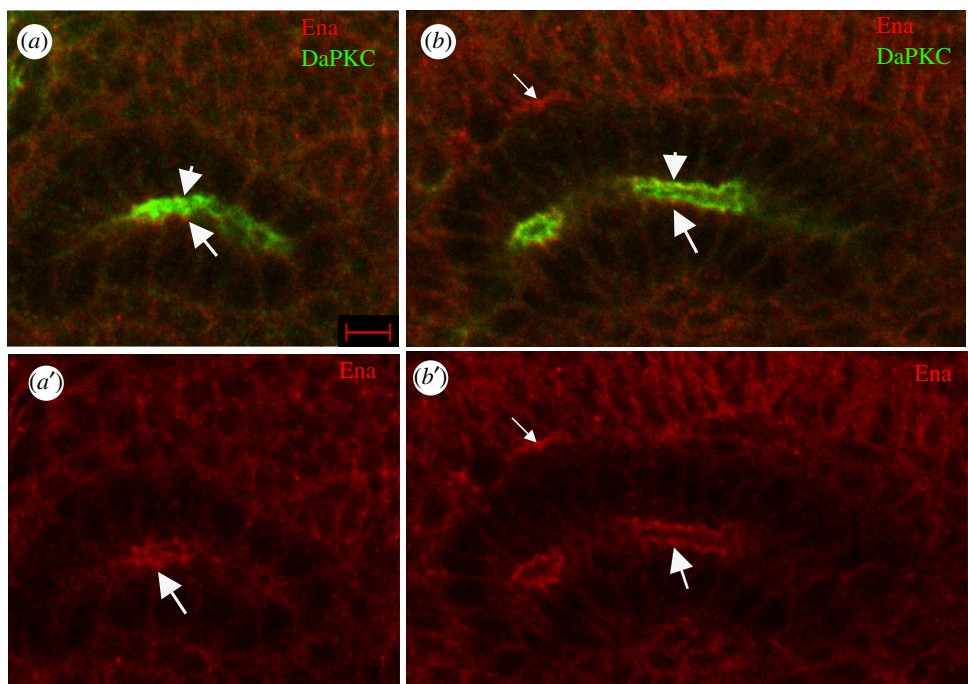

**Figure 2.** Ena localizes to the apical domain of migrating salivary gland cells. (*a,b*) In wild-type salivary glands, Ena (red, large arrows) localized to the apical domain sub-apical to DaPKC (green, arrowheads) at stage 12 when the gland is starting to migrate (*a*) and at stage 14 (*b*) when the gland has completed its posterior migration. Ena also localized to the basal domain in stage 14 gland cells (*b* and *b′*, small arrows). Embryos were labelled for Ena (red) and DaPKC (green). Scale bar in (*a*) represents 5 μm. Images are shown dorsal up and posterior right.

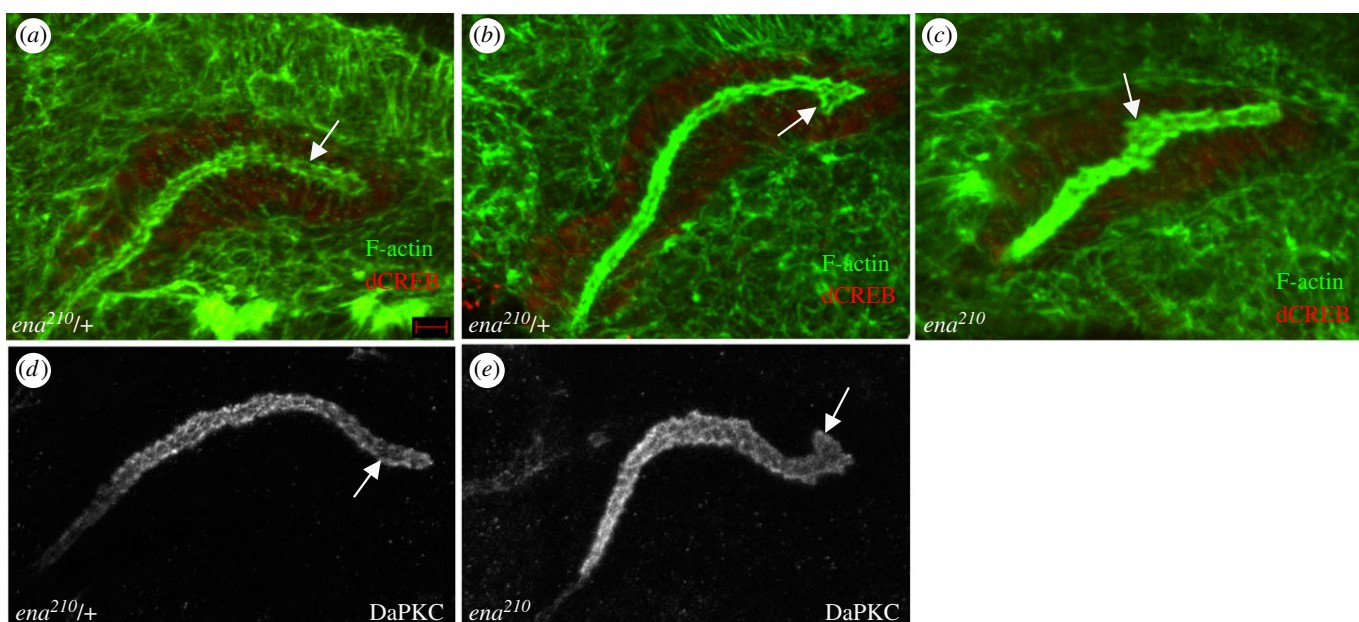

**Figure 3.** Lumen shape is irregular in *ena* mutant salivary gland cells. In *ena²¹⁰* heterozygous embryos, the salivary gland lumen is uniformly shaped (*a,d*, arrows) whereas in *ena²¹⁰* homozygous embryos, the lumen is expanded at the distal tip (*b,e*, arrows) or in the medial region (*c*, arrow). All embryos shown are at stage 13 and were labelled for F-actin (*a–c*, green) and dCREB (*a–c*, red) or DaPKC (*d,e*). Scale bar in (*a*) represents 5 μm. Images are shown dorsal up and posterior right.

## 2.3. Cell division makes a partial contribution to cell translocation during regeneration

To test the universality of the above findings, we turned to another experimental model where cells change their location. Larval imaginal discs develop into adult structures during metamorphosis. The larval wing disc is a single layer of columnar epithelium covered with a layer of squamous cells. The wing disc can regenerate and develop into a normal wing after many types of damage including genetic and surgical ablation of parts of the disc or doses of IR that kill about half of the cells [23–25]. This regeneration occurs without a dedicated stem cell pool. We reported previously that the future wing hinge region of the wing disc shows regenerative properties. Specifically, hinge cells are resistant to killing by IR and can translocate to the wing pouch area that suffers more IR-induced apoptosis, where they convert from hinge to pouch fate and participate in regeneration [12]. This behaviour of the hinge cells was also seen after genetic ablation of the pouch [26].

royalsocietypublishing.org/journal/rsob    Open Biol. 9: 180245

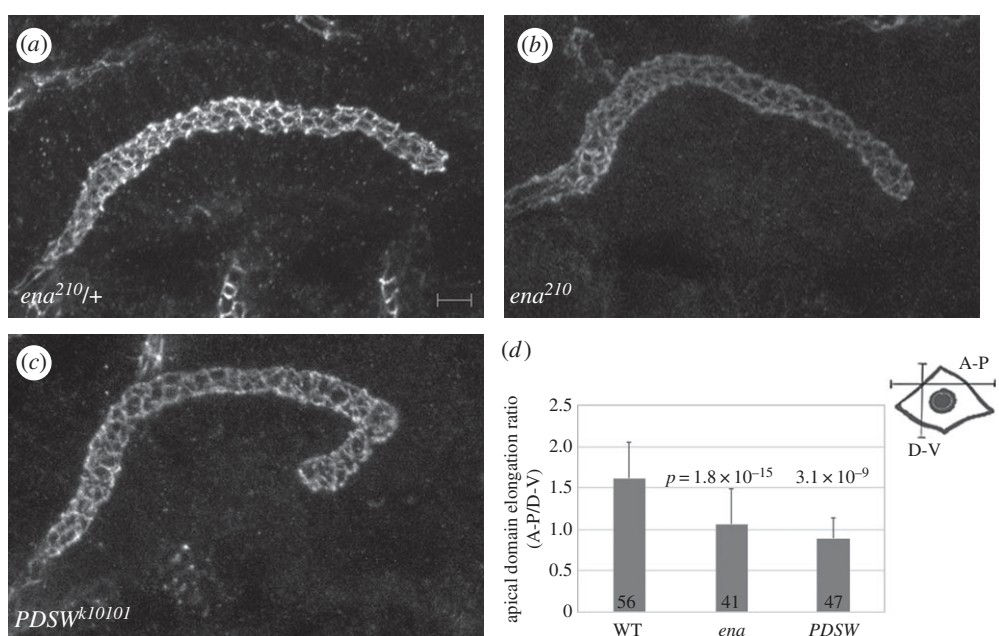

**Figure 4.** Ena and PDSW control apical domain elongation but not size. (*a*–*c*) Immunostaining for DE-cadherin detects the apical domain in *ena*[210] heterozygous (*a*) and homozygous embryos (*b*), and *PDSW*[k10101] homozygous embryos (*c*). (*d*) Graph depicting the apical domain elongation ratio in wild-type, *ena*[210] and *PDSW*[k10101] salivary gland cells. The number of salivary gland cells analysed for apical domain elongation for each genotype is indicated. *p*-values against wild-type controls were calculated using a two-tailed *t*-test. All embryos shown are at stage 14. *ena*[210] embryos were immunostained for β-galactosidase (not shown) to distinguish heterozygous from homozygous embryos. Scale bar in (*a*) represents 5 μm. Images are shown dorsal up and posterior right.

In our published studies, we monitored the translocation of the hinge cells into the pouch and fate change using a G-trace lineage tracing system [27]. Here, GAL4 drives UAS-RFP (real time marker) and UAS-FLP that catalyses a recombination event to result in stable GFP expression (lineage marker). We used the 30A-GAL4 driver to express G-trace in the hinge as we have done in previous studies (figure 5*a*–*c*) [12,13,28]. In un-irradiated discs, red fluorescent protein (RFP) (figure 5*a*) and GFP (figure 5*b*) mostly overlap (figure 5*c*), indicating that cell fates are stable. We have shown before that at 72 h after irradiation with 4000R (40 Gy) of X-rays, GFP+RFP− cells are found in the pouch area (figure 5*d*, enclosed by yellow dashed line) [12]. These are former hinge cells that are expressing the GFP lineage tracer but have lost the hinge fate (became RFP−) and translocated towards the pouch. These cells express the pouch marker VgQ-lacZ [28].

The RFP−GFP+ area enclosed by the yellow line is quantified in Image J [29] and normalized to the RFP+GFP+ area, as a quantitative measure of cell translocation (figure 6*a*, 'GAL4 only'). Expression of UAS-Axin (Wg inhibitor) reveals a cell-autonomous requirement for Wg signalling in the translocating cells ([12; reproduced in new experiments in figure 5*e,f*, quantified in figure 6*a*]). Because Wg and other genes we study are essential for development, we used GAL80[ts] to repress GAL4 and allow disc development to proceed until mid-third instar larva stage (figure 5*m* and Material and methods; see also [12]). GAL4 was de-repressed by a shift to 29°C for 24 h before irradiation and irradiated wing discs analysed 72 h after irradiation. This system has been used successfully to identify genes that regulate regenerative behaviour in the hinge cells including Wg, STAT92E, STAT effector and transcription factor Zfh2, and a member of the nucleosome remodelling complex Nurf-38 [12,13,28]. We used this published assay to investigate the mechanism(s) responsible for cell translocation during regeneration.

Directed cell divisions can play a role in the final placement of cells within a tissue. During *Drosophila* wing development, oriented cell divisions are thought to contribute to the oblong shape of the wing discs and the adult wing because randomizing of division orientation produces a rounded wing [30], although compensatory mechanisms such as cell rearrangements also operate [31]. During cell competition in the wing disc when two groups of cells with different growth characteristics become juxtaposed, dividing 'winner' cells orient their mitotic spindle in such a way that their daughter cells end up among the 'losers' [32]. The hinge cells could likewise direct their daughter cells towards the pouch, and it may be these directed cell divisions that cause cell translocation during regeneration. To address this possibility, we blocked cell divisions in the hinge, by expressing Rux, an inhibitor of mitotic Cdk1 activity. We have used this approach before for a different purpose: to keep the number of hinge cells constant while we monitor an abnormal mode of regeneration that produces an ectopic wing disc [13]. Expression of Rux with the same 30A-GAL4 driver, we showed, kept the cell number constant, i.e. inhibited mitotic divisions. Here, we asked whether the same experimental manipulation prevents the translocation of hinge cells into the pouch. Discs expressing 30A-GAL4 > UAS-*Rux* in the hinge cells show RFP+ cells with enlarged nuclei (figure 5*g*, compare arrow and arrowhead); this is expected as inhibition of Cdk1 blocks mitosis but still allows repeated rounds of S phase [13,33,34]. We found that the hinge cells still translocated into the pouch after irradiation, albeit less efficiently (figure 5*g*; 'Rux' in figure 6*a,b*). Rux expression reduced translocation by about threefold (15 ± 10% from 53 ± 18% in GAL4 only controls). Translocated GFP+RFP− cells show reduced nuclear size, which is expected as cells lose their hinge identity and GAL4/Rux expression (arrow in figure 5*g*). The finding that Rux reduced but did not eliminate cell translocation suggested that we cannot explain all cell translocation by directed divisions. Instead, cell migration may contribute.

royalsocietypublishing.org/journal/rsob    *Open Biol.* **9**: 180245

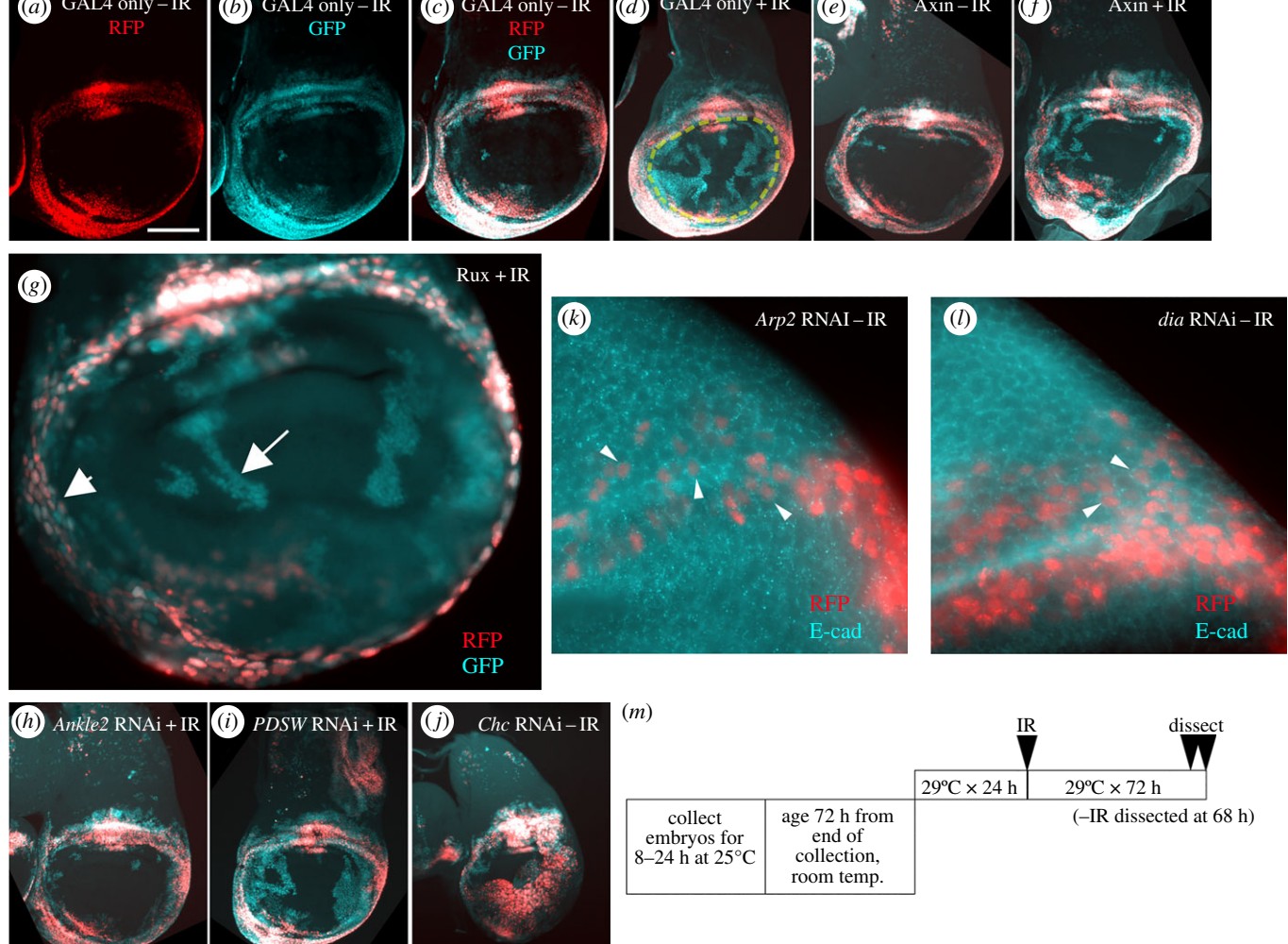

**Figure 5.** Lineage tracing reveals cell translocation during regeneration in irradiated wing discs. Wing discs were fixed after irradiation with 0R (−IR) or 4000R (+IR) of X-rays and visualized for RFP/GFP. Scale bar in (*a*) represents 100 μm in (*a*–*j*), 10 μm in (*k*,*l*) and 25 μm in (*g*). Images are shown dorsal up and posterior right. (*a*–*d*) 30A-GAL4 > UAS-G-trace controls show translocated GFP+RFP− cells after irradiation (within the yellow circle in *d*). (*e*,*f*) Co-expression of Wg inhibitor Axin has been shown before to disrupt cell translocation; 30A-GAL4 > UAS-Axin discs from new experiments are included as positive controls. (*g*) 30A-GAL4 > UAS-Rux disc showing RFP-GFP+ cells in the pouch (arrow). Note the difference in nuclear size from the hinge cells (arrowhead). (*h*,*i*) Irradiated discs from *Ankle2* and *PDSW* RNAi larvae illustrate the range of translocation defects seen in the RNAi lines tested. (*j*) An un-irradiated disc from *Clatherin heavy chain* RNAi larvae shows disc defects. (*k*,*l*) DE-cadherin antibody-stained discs expressing RNAi against *Arp2* (*k*) or *dia* (*l*) illustrate the lack of cytokinesis failure; we did not observe bi- or multi-nucleate cells among those expressing RNAi (RFP+, arrowheads). DE-cadherin antibody marks cell boundaries. (*m*) The temperature shift protocol used to control GAL4.

## 2.4. RNAi against Rac2, MAPk-AK2 and RdgA disrupt cell translocation during regeneration

To address the possibility that cell migration contributes to hinge-to-pouch translocation during regeneration, we tested the effect of some of the RNAi lines used in the salivary gland assay.

From 30 RNAi lines tested in the salivary gland assay, we randomly selected 14 lines to test in the wing discs (three examples shown in figure 5*h*–*j*). Ena and Rab35 were not among the 14 randomly selected. But because we detect a role for these genes in the salivary gland assay (this report), we analysed RNAi against Ena and Rab35 in the wing disc assay. Of the total 16 thus analysed in the wing disc, one produced deformed discs even without irradiation and was not considered further (*Clatherin heavy chain*; figure 5*j*). We also analysed two empty vector controls (KK and GD) and found that the extent of translocation was lower in the GD discs relative to the KK discs although the difference was not statistically significant ($p = 0.153$, two-tailed $t$-test). Nonetheless, for statistical analysis, we compared each RNAi line with its corresponding vector control, KK or GD. Most (14/16) RNAi lines thus analysed showed statistically significant ($p < 0.05$) deviation from the vector controls in the translocation of hinge cells into the pouch but differed in the magnitude of the defect (figure 6). The three strongest hits targeted Ankyrin repeat and LEM domain containing 2 (*Ankle2*, CG8465), *past1* (CG6148) and an uncharacterized gene (CG2794). Ankle2 is required for *Drosophila* S2 cell spreading [14]. It shows high expression in the *Drosophila* larval central nervous system and *Ankle2* mutant larvae show smaller brains than control larvae [35]. Past1 (Putative Achaete Scute Target 1) is a plasma membrane-associated protein that is implicated in endocytosis and genetically interacts with Notch [36–38]. CG2794 encodes an essential gene of unknown function (FBgn0031265 from FB2018_02, released 3 April 2018). *Ankle2*, *past1* and CG2794 have not been implicated in cell movement or regeneration, thus representing novel regulators of these processes.

royalsocietypublishing.org/journal/rsob Open Biol. 9: 180245

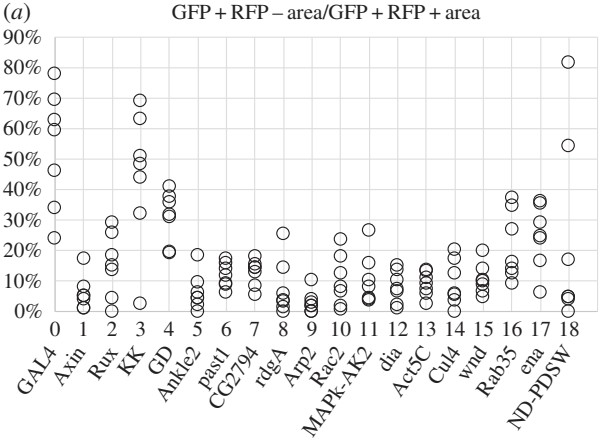

(a) GFP + RFP − area/GFP + RFP + area

(b)

| | VDRC stock# | CG | gene targeted | known/predicted molecular function | p value | p value vs. | hit in salivary gland screen |
|---|---|---|---|---|---|---|---|
| 0 | N/A | N/A | GAL4 only control | N/A | N/A | N/A | N/A |
| 1 | N/A | CG7926 | UAS-Axin (positive control) | Wg inhibitor | 4.582E-05 | GAL4only | N/A |
| 2 | N/A | CG4336 | UAS-Rux | inhibits mitotic cdk | 6.345E-04 | GAL4only | N/A |
| 3 | N/A | N/A | KK vector control | N/A | N/A | N/A | N/A |
| 4 | N/A | N/A | GD vector control | N/A | N/A | N/A | N/A |
| 5 | 24107 | CG8465 | Ankle2 | unknown | 5.0805E-05 | GD | |
| 6 | 22253 | CG6148 | past1 | calcium/GTP-binding | 1.9290E-04 | GD | |
| 7 | 13369 | CG2794 | ... | unknown | 3.1018E-04 | GD | |
| 8 | 28557 | CG42667 | rdgA | diacylglycerol kinase | 3.6773E-04 | GD | YES |
| 9 | 101999 | CG9901 | Arp2 | Actin cytoskeleton | 3.7141E-04 | KK | |
| 10 | 28926 | CG8556 | Rac2 | GTPase | 6.5855E-04 | GD | YES |
| 11 | 3171 | CG3086 | MAPk-AK2 | kinase | 7.1175E-04 | GD | YES |
| 12 | 103914 | CG1768 | dia | actin binding | 1.1674E-03 | KK | |
| 13 | 101438 | CG4027 | Act5C | actin cytoskeleton | 1.2823E-03 | KK | |
| 14 | 105668 | CG8711 | Cul4 | Ub ligase binding | 1.8490E-03 | KK | |
| 15 | 103410 | CG8789 | wnd | MAP kinase kinase kinase | 1.9749E-03 | KK | |
| 16 | 101363 | CG9575 | Rab35 | vesicle transport | 3.1296E-02 | KK | YES |
| 17 | 106484 | CG15112 | ena | actin cytoskeleton | NS 0.054 | KK | YES |
| 18 | 106095 | CG8844 | ND-PDSW | NADH dehydrogenase | NS 0.183 | KK | YES |

**Figure 6.** Quantification of cell movements during regeneration after IR. (a) The GFP + RFP − area in the pouch (inside the yellow circle in figure 5d) was quantified in ImageJ and expressed as a percentage of the total GFP + RFP − area in the hinge in each disc. The data are from +IR samples, n = 7 per genotype. (b) The p-values for the dataset in (a) were calculated using a two-tailed t-test and are shown along with stock information.

The wing disc assay identified two regulators of the actin cytoskeleton (Arp2, dia) and actin itself (Act5; figure 6). The requirement for actin and its regulators in cell translocation could be due to their direct contribution to cell migration or because actin function is required for the cytokinesis step of cell division, which we know from Rux experiments makes a contribution in this experimental model. Examination of Arp2 or dia RNAi discs, however, did not reveal any evidence of cytokinesis failure (figure 5k,l); RFP+ cells that experienced RNAi show discrete nuclei and there is no evidence of bi- or multi-nucleate cells expected from cytokinesis failure (arrowheads). Therefore, we conclude that the role of actin and its regulator in the translocation of hinge cells into the pouch is more likely to be due to their role in cell migration.

## 3. Discussion

In this study, we compared the requirement for collective cell migration during salivary gland development and for cell movements during regeneration of irradiated wing discs, both in Drosophila. Of 30 RNAi lines tested in the salivary glands, eight produced defects and included three known (Rac2, Rab35 and Rab40) and four novel regulators. The eighth is Cdk9, whose effects may be indirect. From all the RNAi lines tested in wing disc regeneration, all except two produced statistically significant effects although the magnitude of the effect varied from line to line. The two exceptions are ena and PDSW, which are precisely two of the eight identified in

salivary gland experiments. Furthermore, the top hits in the wing disc such as Ankle2 and past1 produced no effect in the salivary glands. And the effect of depleting Rab35, a hit in the salivary gland screen, was the weakest among the lines with statistical significance. One possible reason why some of the RNAi lines produced a phenotype in the wing disc but not in the embryonic salivary glands is that the gene product is maternally deposited and the RNAi could not counteract this pool of mRNA. Publicly available data, however, do not support this explanation. Ankle2, for example, is maternally deposited at 111 RPKM (Reads Per Kilobase of transcript, per Million mapped reads) in 0–2-hour-old embryos and its levels declined to 45 RPKM in 8–10-hour-old embryos (stage 12), the relevant stage for salivary gland formation ('gene_rpkm_report_fb_2017_05.tsv' downloaded from Flybase on 03/29/2019). The corresponding values in the same dataset for PDSW were 131 and 121 RPKM, respectively. The values for MAPk-AK2 were 92 and 57 RPKM, respectively. Yet, PDSW and MAPk-AK2 RNAi produced salivary gland defects while Ankle2 RNAi did not.

While these results suggest that cell movements during salivary gland development and during wing disc regeneration after IR damage depend on largely distinct genetic requirements, the data also suggest commonalities. For example, three genes identified in the salivary gland experiments (rdgA, Rac2 and MAPk-AK2) did show defects in the wing disc with p-values in the $10^{-4}$ range. Collectively, these data indicate that cell movements during salivary gland development and during wing disc regeneration after IR damage have common as well as distinct genetic requirements.

One possible mechanism by which MAPk-AK2, RdgA and Rac2 control salivary gland migration and make a partial contribution to cell movements in the wing disc is through regulation of the actin cytoskeleton. We showed previously that known actin regulators, Rac and Rho GTPases, play essential roles in salivary gland migration and that differential distribution of F-actin is important for gland migration and lumen size control [5–8]. RdgA has been shown to regulate fibroblast migration by reorganizing the actin cytoskeleton through activation of Rac1 and Pak1, which dissociates Rac1 from RhoGDI [39]. In migrating endothelial cells MAPk-AK2 activates LIM-kinase 1 to remodel actin through phosphorylation and inactivation of cofilin [40].

The role of Ena and PDSW in the salivary gland could likewise be through the actin cytoskeleton. Ena and its mammalian orthologue, VASP, have been shown in a number of experimental systems to promote actin-based membrane protrusions and cell motility [41,42]. Ena/VASP promotes G-actin incorporation at the growing ends of actin filaments, although the mechanism is still unclear [43]. Although no direct evidence exists for NADH and mitochondrial proteins in regulation of the actin cytoskeleton, interactions between mitochondria and actin have been documented [44]. Regardless of the mechanism, contributions of Ena and PDSW appear less important in the wing disc. By contrast, RNAi against two known regulators of actin, Dia and Arp2, as well as actin (Act5C), produced partial defects in wing disc regeneration but no effect on salivary gland migration. We conclude that, while actin function is important in both systems, how it is regulated may be different.

Of four RNAi lines found to affect salivary gland migration (ena, PDSW, rdgA and MAPk-AK2), only the one

against *ena* disrupted cell rearrangements in the proximal gland. Thus, *ena* may mediate proximal cell rearrangement through Rho GTPase-dependent processes that are distinct from the mechanisms by which MAPk-AK2, RdgA and PDSW mediate actin-dependent gland migration in general. Cell rearrangements appeared normal after RNAi of the latter three genes, yet proximal glands showed defective migration, suggesting that additional steps are required. Ena, as well as the other three genes, may contribute to these additional steps. We envision multiple mechanisms by which Ena, MAPk-AK2 and RdgA could control salivary gland migration, which are not exclusive of each other. Gland migration has been shown to rely on actin-dependent basal membrane protrusions in the distal gland cells and cell shape changes in the proximal gland cells [6,7]. Actin-dependent integrin localization and/or function in the surrounding mesoderm-derived tissues and/or the gland cells could also be another mechanism for control of gland migration [2,6]. Additional studies are necessary to determine if one or more of these mechanisms are responsible for the observed gland migration defects.

Of the four genes analysed in more detail, depletion of only *ena* and *PDSW* showed lumen defects. In *ena* and *PDSW* mutant gland cells, the apical domain area is comparable to that of wild-type cells; however, the apical domains failed to elongate in the direction of migration. With the present data, we cannot distinguish whether the failure to elongate the apical domain is the cause or the consequence of defective gland migration. One way to address possibilities would be with separation-of-function alleles. A mutant that shows normal apical domain elongation but defective posterior migration would suggest that elongation defects are not a consequence of migration defects. Regardless, lumen defects distinguish *ena* and *PDSW* from *rdgA* and *MAPk-AK2* in salivary gland morphogenesis.

Two actin regulators, *Arp2* and *dia*, are among those found in the RNAi experiments. Arp2/3 complex functions widely among eukaryotes to nucleate and organize the actin cytoskeleton [45]. In *Drosophila*, it is required for the formation of the ring canal in the ovary and for the organization of the parallel actin bundles in developing bristles [46], for wound closure [47,48] and for lamella formation in S2 cells [14], among others. Arp2/3 has not been implicated in cytokinesis, and germline cell divisions in the mutants are normal [46]. Dia binds to F-actin and facilitates its contact with the cell membrane. *Drosophila dia* mutants do show cytokinesis defects to produce multi-nucleate spermatids, polyploid larval neuroblasts and polyploid adult follicle cells [49]. But we did not detect cytokinesis defects in *dia* RNAi discs. This could be because of a redundant function or because the depletion, although strong enough to affect cell movement, was not strong enough to prevent cytokinesis. Despite these uncertainties, our data collectively indicate that cell movements during regeneration have contributions from both cell division (Rux experiments) and from cell migration (RNAi experiments). Each of these is complex processes that rely on a very large number of cytoskeletal components and their regulators. This, we propose, may be why a larger fraction of the RNAi lines tested showed a (partial) defect in the wing disc system than in the salivary glands. Understanding how each gene identified in this work contributes to cell movement during normal development and regeneration will be a future goal.

# 4. Material and methods

## 4.1. *Drosophila* techniques

For the analysis of salivary glands, Canton-S flies were used as wild-type controls. *ND-PDSW*[k10101], *ena*[02029], *ena*[210], ena[23], *MAPk-AK2*[G265], *rdgA*[3] and UAS-GFP-NLS were obtained from the Bloomington Stock Center and are described in FlyBase (http://flybase.bio.indiana.edu/). For salivary gland-specific expression of the UAS constructs, we used *fork head* (*fkh*)-GAL4. The UAS-GFP-NLS *fkh*-GAL4 recombinant line was generated using standard genetic techniques.

For the analysis of wing disc, the larvae were raised in Nutri-Fly Bloomington formula food (Genesee Scientific) at room temperature unless otherwise noted. The embryos were collected for 8–24 h at 25°C and reared at room temperature for 72 h from the end of the collection. The vials were monitored regularly for overcrowding (typically seen as dimples in the food) and split to prevent overcrowding. The larvae were then shifted to 29°C for 24 h before irradiation to de-repress GAL4. The un-irradiated controls remained at 29°C for an additional 38–48 h while the irradiated samples were kept at 29°C for 72 h post-irradiation. For balanced RNAi lines, the absence of CyO balancer-encoded GFP was used to identify the experimental animals.

## 4.2. Irradiation

The larvae in food were placed in Petri dishes and irradiated in a Faxitron Cabinet X-ray System Model RX-650 (Lincolnshire, IL) at 115 kV and 5.33 rad s$^{-1}$.

## 4.3. Immunocytochemistry and *in situ* hybridization

Embryo fixation and antibody staining were performed as previously described [7]. The following antisera were used at the indicated dilutions: rat dCREB antiserum at 1:10 000; rat antisera to DE-cadherin at 1:400 (Developmental Studies Hybridoma Bank, DSHB; Iowa City, IA); rabbit aPKC antiserum (Santa Cruz Biotechnology, Dallas, TX) at 1:500; mouse β-galactosidase (β-gal) antiserum (Promega, Madison, WI) at 1:500. Appropriate AlexaFluor 488-, 647- or rhodamine- (Molecular Probes, Eugene, OR) conjugated secondary antibodies were used at a dilution of 1:500 for salivary glands. Anti-mouse-Cy5 secondary antibody was used at 1:200 dilution for wing discs (Jackson, West Grove, PA). Stained embryos were mounted in Aqua Polymount (Polysciences, Inc., Warrington, PA) and thick (1 μm) fluorescence images were acquired on a Zeiss Axioplan microscope (Carl Zeiss) equipped with an LSM 710 for laser scanning confocal microscopy.

To collect wing discs, the larvae were dissected in 1× phosphate-buffered saline (PBS) and fixed in 4% paraformaldehyde for 30 min at room temperature, washed with 1× PBS for 10 min, followed by a 5 min wash with PBTx (0.1% Triton X-100). The wing discs were stained with 10 μg ml$^{-1}$ Hoechst33342 in PBTx for 2 min, washed three times in PBTx and mounted on glass slides in Fluromount G (SouthernBiotech).

## 4.4. Image analysis

Salivary gland images were acquired on a Zeiss Axioplan microscope (Carl Zeiss) equipped with confocal laser

royalsocietypublishing.org/journal/rsob    Open Biol. **9**: 180245

royalsocietypublishing.org/journal/rsob Open Biol. 9: 180245

scanning microscopy (LSM710, Medgar Evers College-CUNY, New York, NY). The number of nuclei surrounding the central lumen and apical domain area and an elongation ratio of salivary gland cells were measured and quantified as previously described [6,7]. Statistical analysis performed in Microsoft Excel using a two-tailed $t$-test.

Wing disc images were taken on a Leica DMR compound fluorescence microscope using a Q-Imaging R6 CCD camera and Ocular software. The images were assembled, processed and quantified using ImageJ software. Figure 5$j$,$k$ (100× images) were taken on a PerkinElmer spinning disc confocal attached to a Nikon inverted microscope, using an SDS aAndor iXon Ultra (DU-897) EM CCD camera. For wing discs, we dissected a minimum of 10 larvae per genotype per condition and mounted all discs onto a slide. We then randomly selected and imaged seven intact, flat discs for quantification. To justify the sample size of seven, we used a simplified resource equation from [50]. Briefly, $E$ = total number of animals − total number of groups, where an $E$ value of 10–20 is considered adequate. When we compared two groups (vector versus RNAi, for example), six per group or $E$ = 11 would be adequate. Therefore, we chose seven.

Data accessibility. This article does not contain any additional data
Competing interests. We declare we have no competing interests.
Funding. Transgenic RNAi fly stocks were obtained from the Vienna Drosophila Resource Center (VDRC, www.vdrc.at) [51]. S.V. and T.T.S. were supported by NIH grants R01 GM106317 and R35 GM130374, both to T.T.S. M.L. was supported by the Biological Sciences Initiative of the University of Colorado at Boulder. M.M.M., D.L., A.M. and J.M.L. were supported by a Research Initiative for Scientific Enhancement (grant no. R25 GM105553) to Medgar Evers College.

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
