## [Reviewer comments · Open Biology]

Review History

RSOB-18-0245.R0 (Original submission)

Review form: Reviewer 1

Recommendation

Reject – article is scientifically unsound

Are each of the following suitable for general readers?

- a) **Title**
No

- b) **Summary**
Yes

c) **Introduction**

Yes

Is the length of the paper justified?

Yes

Should the paper be seen by a specialist statistical reviewer?

No

Is it clear how to make all supporting data available?

Yes

Is the supplementary material necessary; and if so is it adequate and clear?

Yes

Do you have any ethical concerns with this paper?

No

Comments to the Author

Myat and colleagues screen 30 RNAi lines for salivary gland migration defects and 15 lines for cell translocation defects after wing disc irradiation.

The work is neither a screen nor an investigation of a developmental event or gene. With such a tiny selection of RNAi lines one cannot judge if the reported mild phenotypes are real or not (especially when 13 out of 15 show a phenotype as in the irradiation experiment). In addition, there are technical issues, e.g. there is no quantification for the salivary gland screen data, and for the wing disc screen 2 wing discs from the same larva appear to have been used, further reducing the statistical significance (down to 4 independent data points). I would believe these hits only if a random selection of 30 or 15 RNAi lines results in 0-1 hits in the salivary gland and wing disc assays. 20-87% hits out of the total is not a screen.

Minor comments:

I cannot understand how these 30 RNAi lines were chosen. A rationale should be provided (e.g. line 1 had phenotype x in screen y) in an extra column. Also, why many other hits from the referenced screens were not chosen.

Line 125: Supplemental Table 1 should be referenced

Line 132: here and throughout the manuscript, gene names should be consistent and correspond to official FlyBase nomenclature, e.g. DGK is *rdgA*, *pds* is ND-PDSW, *enabled* is *ena* etc.

Line 184: what about MAPk-AK2 apical domain defects?

Line 388: ...is still oriented dorsally...

Line 441: are we talking about n=7 larvae or n=7 wing discs from 4 larvae? One would expect very similar results for 2 wing discs coming from the same larva.

There is quite a bit of discussion in the last part of the results section.

Supplemental table 1:

It would make more sense to have one column labeled symbol, and give in this column either the gene symbol or the CG number, then in column 2 the putative molecular function, e.g. diacylglycerol kinase or actin, etc.

As it is right now, column 2 sometimes gives the gene name, sometimes the symbol, sometimes a mix.

Review form: Reviewer 2

Recommendation

Major revision is needed (please make suggestions in comments)

Are each of the following suitable for general readers?

- a) **Title**
Yes
- b) **Summary**
Yes
- c) **Introduction**
Yes

Is the length of the paper justified?

Yes

Should the paper be seen by a specialist statistical reviewer?

No

Is it clear how to make all supporting data available?

Yes

Is the supplementary material necessary; and if so is it adequate and clear?

Yes

Do you have any ethical concerns with this paper?

No

Comments to the Author

In this work, Myat et al. use different *Drosophila* models to investigate the contribution of selected genes with potential roles in collective cell migration. On the one hand they use the embryonic salivary gland, a well-established model, and on the other they analyze cell repositioning in wing disc regeneration. Their experiments shed light into the mechanism of this type of cell position change during regeneration. In addition they identify new genetic requirements for migration in salivary glands and in disc regeneration. A small part of the genes tested are shown to be required for migration in both systems. They conclude that different types of cell movements have common and specific genetic requirements.

The analysis presented allows the identification of new players in salivary gland formation and in regeneration. However, besides this descriptive (phenotypic) analysis, no molecular mechanisms are provided or presented.

Here I suggest different aspects that I consider that could help to improve the manuscript

1. Ena activity.

The authors show that in *ena* mutants the distal part of the gland turns posteriorly but the proximal part does not. They observe lack of cell rearrangements in the proximal part. It is known that salivary gland distal cells extend actin-based protrusions. As the authors mention, it is also known that Ena can promote actin-based protrusions. These evidences raise the hypothesis that Ena promotes actin protrusions at the distal region and that this promotes proper

migration/pulling force of the "leading" (distal) region that could indirectly promote cell rearrangements in the proximal part.

Can the authors analyze actin protrusions in Ena mutants and compare to the wild type?

2. Ena, MAPK-AK2, DGK and pdsw produce a similar salivary gland phenotype, however, at the cellular level (cell rearrangements) the effects seem quite different.

Could the authors elaborate on whether they consider that these genes belong to 2 different mechanisms? Would MAPK-AK2, DGK and pdsw participate in a single mechanism?

3. Ena localises apically and is enriched in some basal domains. In Ena mutants the authors show defects of lumen shape. They find that apical area is maintained but cells are not elongated along the proximo-distal axis as in the wild type. This effect is also observed in pdsw downregulation, although the mechanisms may be different.

Do they also detect a similar effect downregulating MAPK-AK2, DGK?

Are tubes, in general, shorter and wider?

4. The authors report defects in cell shape in Ena and in pdsw mutants. Could these defects underlie the phenotype of salivary glands observed, rather than being a consequence of impaired collective cell migration? How can the authors distinguish between defects in migration or defects in cell shape during salivary gland migration?

5. I find the results section about wing disc regeneration a bit confusing.

I suggest to divide the chapter into a first one where they show that cell repositioning relies on cell migration as well as directed divisions. This is a very interesting result by itself and helps to interpret the rest of results. Another chapter/s could document the effect of downregulation of the different genes tested in cell repositioning. I also feel that some aspects in this results section could be transferred to the discussion section.

6. Surprisingly, from all lines tested in wing disc regeneration, all except 2 produce an effect. These lines are precisely two of the ones identified in salivary gland experiments. In addition, the rest of lines required for salivary gland formation produce mild effects in wing disc regeneration. Furthermore, the top hits in wing disc produce no effects in salivary glands. Wouldn't these results suggest that cell movements during salivary gland development and during wing disc regeneration after IR damage depend on rather distinct genetic requirements?

Decision letter (RSOB-18-0245.R0)

02-Jan-2019

Dear Dr Su:

We are writing to inform you that the Editor has reached a decision on your manuscript RSOB-18-0245 entitled "Directed RNAi screens identify novel regulators of cell movement in development and regeneration in *Drosophila*", submitted to Open Biology.

As you will see from the reviewers' comments below, there are a number of criticisms that prevent us from accepting your manuscript at this stage. The reviewers suggest, however, that a revised version could be acceptable, if you are able to address their concerns. If you think that you can deal satisfactorily with the reviewer's suggestions, we would be pleased to consider a revised manuscript.

The revision will be re-reviewed, where possible, by the original referees. As such, please submit the revised version of your manuscript within six weeks. If you do not think you will be able to meet this date please let us know immediately.

When submitting your revised manuscript, please respond to the comments made by the referee(s) and upload a file "Response to Referees" in "Section 6 - File Upload". You can use this to document any changes you make to the original manuscript. In order to expedite the processing of the revised manuscript, please be as specific as possible in your response to the referee(s).

Please see our detailed instructions for revision requirements
<https://royalsociety.org/journals/authors/author-guidelines/>

Sincerely,

The Open Biology Team
mailto: openbiology@royalsociety.org

ditage Insights by clicking on the following link: <https://www.surveymonkey.com/r/author-perspectives-on-academic-publishing-royal-society>

This should take no more than 15 minutes and you will have the opportunity to enter a prize draw. We hope these results will provide us with valuable insights we can use to improve our service.

Editor's Comments to Author(s):

Reviewer(s)' Comments to Author(s):

Referee: 1

Comments to the Author(s)

Myat and colleagues screen 30 RNAi lines for salivary gland migration defects and 15 lines for cell translocation defects after wing disc irradiation.

The work is neither a screen nor an investigation of a developmental event or gene. With such a tiny selection of RNAi lines one cannot judge if the reported mild phenotypes are real or not (especially when 13 out of 15 show a phenotype as in the irradiation experiment). In addition, there are technical issues, e.g. there is no quantification for the salivary gland screen data, and for the wing disc screen 2 wing discs from the same larva appear to have been used, further reducing the statistical significance (down to 4 independent data points). I would believe these hits only if

a random selection of 30 or 15 RNAi lines results in 0-1 hits in the salivary gland and wing disc assays. 20-87% hits out of the total is not a screen.

Minor comments:

I cannot understand how these 30 RNAi lines were chosen. A rationale should be provided (e.g. line 1 had phenotype x in screen y) in an extra column. Also, why many other hits from the referenced screens were not chosen.

Line 125: Supplemental Table 1 should be referenced

Line 132: here and throughout the manuscript, gene names should be consistent and correspond to official FlyBase nomenclature, e.g DGK is *rdgA*, *pds* is ND-PDSW, *ena* is *ena* etc.

Line 184: what about MAPK-AK2 apical domain defects?

Line 388: ...is still oriented dorsally...

Line 441: are we talking about n=7 larvae or n=7 wing discs from 4 larvae? One would expect very similar results for 2 wing discs coming from the same larva.

There is quite a bit of discussion in the last part of the results section.

Supplemental table 1:

It would make more sense to have one column labeled symbol, and give in this column either the gene symbol or the CG number, then in column 2 the putative molecular function, e.g. diacylglycerol kinase or actin, etc.

As it is right now, column 2 sometimes gives the gene name, sometimes the symbol, sometimes a mix.

Referee: 2

Comments to the Author(s)

In this work, Myat et al. use different *Drosophila* models to investigate the contribution of selected genes with potential roles in collective cell migration. On the one hand they use the embryonic salivary gland, a well-established model, and on the other they analyze cell repositioning in wing disc regeneration. Their experiments shed light into the mechanism of this type of cell position change during regeneration. In addition they identify new genetic requirements for migration in salivary glands and in disc regeneration. A small part of the genes tested are shown to be required for migration in both systems. They conclude that different types of cell movements have common and specific genetic requirements.

The analysis presented allows the identification of new players in salivary gland formation and in regeneration. However, besides this descriptive (phenotypic) analysis, no molecular mechanisms are provided or presented.

Here I suggest different aspects that I consider that could help to improve the manuscript

1. Ena activity.

The authors show that in *ena* mutants the distal part of the gland turns posteriorly but the proximal part does not. They observe lack of cell rearrangements in the proximal part.

It is known that salivary gland distal cells extend actin-based protrusions. As the authors mention, it is also known that Ena can promote actin-based protrusions. These evidences raise the hypothesis that Ena promotes actin protrusions at the distal region and that this promotes proper migration/pulling force of the "leading" (distal) region that could indirectly promote cell rearrangements in the proximal part.

Can the authors analyze actin protrusions in *Ena* mutants and compare to the wild type?

2. Ena, MAPK-AK2, DGK and pdsw produce a similar salivary gland phenotype, however, at the cellular level (cell rearrangements) the effects seem quite different.

Could the authors elaborate on whether they consider that these genes belong to 2 different mechanisms? Would MAPK-AK2, DGK and pdsw participate in a single mechanism?

3. Ena localises apically and is enriched in some basal domains. In Ena mutants the authors show defects of lumen shape. They find that apical area is maintained but cells are not elongated along the proximo-distal axis as in the wild type. This effect is also observed in pdsw downregulation, although the mechanisms may be different.

Do they also detect a similar effect downregulating MAPK-AK2, DGK?

Are tubes, in general, shorter and wider?

4. The authors report defects in cell shape in Ena and in pdsw mutants. Could these defects underlie the phenotype of salivary glands observed, rather than being a consequence of impaired collective cell migration? How can the authors distinguish between defects in migration or defects in cell shape during salivary gland migration?

5. I find the results section about wing disc regeneration a bit confusing.

I suggest to divide the chapter into a first one where they show that cell repositioning relies on cell migration as well as directed divisions. This is a very interesting result by itself and helps to interpret the rest of results. Another chapter/s could document the effect of downregulation of the different genes tested in cell repositioning. I also feel that some aspects in this results section could be transferred to the discussion section.

6. Surprisingly, from all lines tested in wing disc regeneration, all except 2 produce an effect. These lines are precisely two of the ones identified in salivary gland experiments. In addition, the rest of lines required for salivary gland formation produce mild effects in wing disc regeneration. Furthermore, the top hits in wing disc produce no effects in salivary glands. Wouldn't these results suggest that cell movements during salivary gland development and during wing disc regeneration after IR damage depend on rather distinct genetic requirements?

Author's Response to Decision Letter for (RSOB-18-0245.R0)

See Appendix A.

RSOB-18-0245.R1 (Revision)

Review form: Reviewer 1

Recommendation

Accept as is

Are each of the following suitable for general readers?

- a) **Title**
Yes
- b) **Summary**
Yes
- c) **Introduction**
Yes

Is the length of the paper justified?

Yes

Should the paper be seen by a specialist statistical reviewer?

No

Is it clear how to make all supporting data available?

Yes

Is the supplementary material necessary; and if so is it adequate and clear?

Yes

Do you have any ethical concerns with this paper?

No

Comments to the Author

My concerns have been addressed

Review form: Reviewer 2

Recommendation

Accept as is

Are each of the following suitable for general readers?

- a) **Title**
Yes
- b) **Summary**
Yes
- c) **Introduction**
Yes

Is the length of the paper justified?

Yes

Should the paper be seen by a specialist statistical reviewer?

No

Is it clear how to make all supporting data available?

Yes

Is the supplementary material necessary; and if so is it adequate and clear?

Yes

Do you have any ethical concerns with this paper?

No

Comments to the Author

Myat et al. addressed all the points raised by this reviewer. They have answered most of the concerns and have incorporated the suggestion mainly in the discussion section. In this way they have clarified several points.

I consider that the manuscript improved with the changes, although it is still a descriptive work with no clear molecular mechanisms tested. There is one experiment that was asked by this reviewer and that has not been successfully addressed. The authors indicate that although it is an interesting question, the experiment cannot be carried out due to the unavailability of experienced researcher.

Review form: Reviewer 3

Recommendation

Major revision is needed (please make suggestions in comments)

Are each of the following suitable for general readers?

- a) **Title**
Yes
- b) **Summary**
Yes
- c) **Introduction**
Yes

Is the length of the paper justified?

Yes

Should the paper be seen by a specialist statistical reviewer?

No

Is it clear how to make all supporting data available?

Yes

Is the supplementary material necessary; and if so is it adequate and clear?

Yes

Do you have any ethical concerns with this paper?

No

Comments to the Author

In this manuscript, Myat et al. used two *Drosophila* systems – embryonic salivary gland and larval wing disc – to better understand the genetic requirements for collective migration. The main strategy was knocking down genes using RNAi lines that have been known to affect cell membrane morphology, cytoskeleton and cell adhesion in either system (30 genes for salivary glands and 16 for wing discs). From the RNAi phenotypes, they conclude that cell movements in development and regeneration have common as well as different genetic requirements.

My biggest concern is the authors' interpretation of RNAi phenotypes in embryonic salivary glands. It is not uncommon that knocking down essential genes zygotically in fly embryos does not provide an overt phenotype due to strong maternal contribution. The authors say that they observed 8 out of 30 RNAi lines show migration defects of salivary glands, but it does not imply that the rest of 22 genes do not have a role in salivary gland migration. It is possible that maternally provided gene products were enough to allow the salivary gland to migrate properly without any overt migration defects. Indeed, high throughput expression data in Flybase shows “high expression” of *Past1*, “very high expression” of *Ankle2* in 0-2 hr embryos. These are the two genes that they mention in Discussion as being the top hits in the wing disc but showing no effect in the salivary gland (line 300). It is also true for *Dia* (moderate expression), *Arp2* (high expression) and *Act5c* (extremely high expression), another three genes that they mention in line 328 as the same cases. I did not look up the expression level of every single gene that the tested in this study, but I suspect that many genes would be the same as most of them are essential genes that have important functions in various cell and developmental contexts. The authors' conclusion on this paper is based on their interpretation of which RNAi lines show migration defects in either system, and I can't agree with it. I strongly suggest that the authors should consider and provide an alternative interpretation on their RNAi data in salivary glands. It is even possible that almost every single gene might have a role in salivary gland migration as it does in their wing disc translocation assay.

They also only tested a single RNAi for each gene, which further raises a concern that one won't be able to make a conclusion whether the absence of the phenotype is because those genes were not required for salivary gland migration or because these RNAi lines did not successfully knock down those genes.

Minor comments:

1. The authors showed mild salivary gland migration defects with 8 genes. This is a quite subtle phenotype and should be described/quantified better.

a. Figure 1. It took me a little while to clearly understand what the posterior migration defects are. The figure legends say arrow and arrowheads in there, but they are not shown in the images.

b. How many salivary glands were analyzed for this phenotype? And what is the penetrance of the phenotype? Both the number of glands and the penetrance of the phenotype should be included. They mentioned that they looked at live embryos first and then confirmed it with fixed samples when they observe more than 5 live samples with abnormal morphology. But it is not clear how many stage 14 salivary glands they quantified for the proximal migration defects.

c. It is my understanding that *Drosophila* embryonic salivary glands migrate actively through stages 13-16. Do these proximal migration defects persist at later stages as well? Or is this something transient that is only observed at st 14? If the latter, could it be a slight delay of migration?

d. Lines 159-162: The cell rearrangement defects are only described in the text but never shown in

the figure. It could help readers understand the defects better if images are provided as additional panels in the figure. I understand that the senior author who is responsible for this part does not have access to the equipment anymore, but it could be done easily if the original confocal data are still available.

2. Ena loss of function phenotype

a. Line 186: No effects on localization of DaPKC and DE-Cad in ena mutants also could be due to maternal contribution.

b. Figure 4. It should be better described how the apical area was measured. They cited previous papers where the same quantification had been done in the Method section, but adding arrows along the A-P and D-V axis in the image (or a cartoon) to show what is measured would help.

3. Directed cell divisions and wing shape

a. Line 236: "randomizing of division orientation produces a round wing (29)"

It is my understanding that the reference 29 showed a correlation between oriented cell division and the wing shape, but did not show the causal relationship. On the contrary, a very recent paper [Zhou et al., (2019) Oriented Cell Divisions Are Not Required for Drosophila Wing Shape. *Curr Biol* 29: 856-864] showed that oriented cell divisions are not required for Drosophila wing shape.

b. Lines 295-296: "Of 30 RNAi lines tested in the salivary glands, eight produced defects and included two known and four novel regulators."

If two are known regulators and four are novel, what are the remaining two out of eight?

Decision letter (RSOB-18-0245.R1)

21-Mar-2019

Dear Dr Su

We are pleased to inform you that your manuscript RSOB-18-0245.R1 entitled "Regulators of cell movement during development and regeneration in Drosophila" has been accepted by the Editor for publication in *Open Biology*. The reviewer(s) have recommended publication, but also suggest some minor revisions to your manuscript. Therefore, we invite you to respond to the reviewer(s)' comments and revise your manuscript.

Please submit the revised version of your manuscript within 14 days. If you do not think you will be able to meet this date please let us know immediately and we can extend this deadline for you.

- 1) A text file of the manuscript (doc, txt, rtf or tex), including the references, tables (including captions) and figure captions. Please remove any tracked changes from the text before submission. PDF files are not an accepted format for the "Main Document".
- 2) A separate electronic file of each figure (tiff, EPS or print-quality PDF preferred). The format should be produced directly from original creation package, or original software format. Please note that PowerPoint files are not accepted.
- 3) Electronic supplementary material: this should be contained in a separate file from the main text and meet our ESM criteria (see <http://royalsocietypublishing.org/instructions-authors#question5>). All supplementary materials accompanying an accepted article will be treated as in their final form. They will be published alongside the paper on the journal website and posted on the online figshare repository. Files on figshare will be made available approximately one week before the accompanying article so that the supplementary material can be attributed a unique DOI.

Online supplementary material will also carry the title and description provided during submission, so please ensure these are accurate and informative. Note that the Royal Society will not edit or typeset supplementary material and it will be hosted as provided. Please ensure that the supplementary material includes the paper details (authors, title, journal name, article DOI). Your article DOI will be 10.1098/rsob.2016[last 4 digits of e.g. 10.1098/rsob.20160049].

- 4) A media summary: a short non-technical summary (up to 100 words) of the key findings/importance of your manuscript. Please try to write in simple English, avoid jargon, explain the importance of the topic, outline the main implications and describe why this topic is newsworthy.

Images

Data-Sharing

It is a condition of publication that data supporting your paper are made available. Data should be made available either in the electronic supplementary material or through an appropriate repository. Details of how to access data should be included in your paper. Please see <http://royalsocietypublishing.org/site/authors/policy.xhtml#question6> for more details.

Data accessibility section

Sincerely,

The Open Biology Team
 mailto:openbiology@royalsociety.org

Editor's comment:

Please try and address the comments of all of the reviewers.

Reviewer(s)' Comments to Author:

Referee: 1

Comments to the Author(s)
 My concerns have been addressed

Referee: 2

Comments to the Author(s)

Myat et al. addressed all the points raised by this reviewer. They have answered most of the concerns and have incorporated the suggestion mainly in the discussion section. In this way they have clarified several points.

I consider that the manuscript improved with the changes, although it is still a descriptive work with no clear molecular mechanisms tested. There is one experiment that was asked by this reviewer and that has not been successfully addressed. The authors indicate that although it is an interesting question, the experiment cannot be carried out due to the unavailability of experienced researcher.

Referee: 3

Comments to the Author(s)

In this manuscript, Myat et al. used two *Drosophila* systems – embryonic salivary gland and larval wing disc – to better understand the genetic requirements for collective migration. The main strategy was knocking down genes using RNAi lines that have been known to affect cell membrane morphology, cytoskeleton and cell adhesion in either system (30 genes for salivary glands and 16 for wing discs). From the RNAi phenotypes, they conclude that cell movements in development and regeneration have common as well as different genetic requirements.

My biggest concern is the authors' interpretation of RNAi phenotypes in embryonic salivary glands. It is not uncommon that knocking down essential genes zygotically in fly embryos does not provide an overt phenotype due to strong maternal contribution. The authors say that they

observed 8 out of 30 RNAi lines show migration defects of salivary glands, but it does not imply that the rest of 22 genes do not have a role in salivary gland migration. It is possible that maternally provided gene products were enough to allow the salivary gland to migrate properly without any overt migration defects. Indeed, high throughput expression data in Flybase shows “high expression” of *Past1*, “very high expression” of *Ankle2* in 0-2 hr embryos. These are the two genes that they mention in Discussion as being the top hits in the wing disc but showing no effect in the salivary gland (line 300). It is also true for *Dia* (moderate expression), *Arp2* (high expression) and *Act5c* (extremely high expression), another three genes that they mention in line 328 as the same cases. I did not look up the expression level of every single gene that the tested in this study, but I suspect that many genes would be the same as most of them are essential genes that have important functions in various cell and developmental contexts. The authors’ conclusion on this paper is based on their interpretation of which RNAi lines show migration defects in either system, and I can’t agree with it. I strongly suggest that the authors should consider and provide an alternative interpretation on their RNAi data in salivary glands. It is even possible that almost every single gene might have a role in salivary gland migration as it does in their wing disc translocation assay.

They also only tested a single RNAi for each gene, which further raises a concern that one won’t be able to make a conclusion whether the absence of the phenotype is because those genes were not required for salivary gland migration or because these RNAi lines did not successfully knock down those genes.

Minor comments:

1. The authors showed mild salivary gland migration defects with 8 genes. This is a quite subtle phenotype and should be described/quantified better.

a. Figure 1. It took me a little while to clearly understand what the posterior migration defects are. The figure legends say arrow and arrowheads in there, but they are not shown in the images.

b. How many salivary glands were analyzed for this phenotype? And what is the penetrance of the phenotype? Both the number of glands and the penetrance of the phenotype should be included. They mentioned that they looked at live embryos first and then confirmed it with fixed samples when they observe more than 5 live samples with abnormal morphology. But it is not clear how many stage 14 salivary glands they quantified for the proximal migration defects.

c. It is my understanding that *Drosophila* embryonic salivary glands migrate actively through stages 13-16. Do these proximal migration defects persist at later stages as well? Or is this something transient that is only observed at st 14? If the latter, could it be a slight delay of migration?

d. Lines 159-162: The cell rearrangement defects are only described in the text but never shown in the figure. It could help readers understand the defects better if images are provided as additional panels in the figure. I understand that the senior author who is responsible for this part does not have access to the equipment anymore, but it could be done easily if the original confocal data are still available.

2. *Ena* loss of function phenotype

a. Line 186: No effects on localization of DaPKC and DE-Cad in *ena* mutants also could be due to maternal contribution.

b. Figure 4. It should be better described how the apical area was measured. They cited previous papers where the same quantification had been done in the Method section, but adding arrows along the A-P and D-V axis in the image (or a cartoon) to show what is measured would help.

3. Directed cell divisions and wing shape

a. Line 236: “randomizing of division orientation produces a round wing (29)”

It is my understanding that the reference 29 showed a correlation between oriented cell division and the wing shape, but did not show the causal relationship. On the contrary, a very recent paper [Zhou et al., (2019) Oriented Cell Divisions Are Not Required for Drosophila Wing Shape. *Curr Biol* 29: 856-864] showed that oriented cell divisions are not required for Drosophila wing shape.

b. Lines 295-296: “Of 30 RNAi lines tested in the salivary glands, eight produced defects and included two known and four novel regulators.”

If two are known regulators and four are novel, what are the remaining two out of eight?

Author's Response to Decision Letter for (RSOB-18-0245.R1)

See Appendix B.

Decision letter (RSOB-18-0245.R2)

05-Apr-2019

Dear Dr Su

We are pleased to inform you that your manuscript entitled "Regulators of cell movement during development and regeneration in Drosophila" has been accepted by the Editor for publication in *Open Biology*.

Thank you for your fine contribution. On behalf of the Editors of *Open Biology*, we look forward to your continued contributions to the journal.

Sincerely,

The Open Biology Team
mailto: openbiology@royalsociety.org

Appendix A

Dear Editors and Referees,

Thank you for your time and your insightful comments. We provide a point-by-point response below and indicate where in the revised text we have made the corresponding changes. Significant changes to the text are marked in the 'highlighted' version that is submitted along with the clean version. Smaller changes such as the harmonization of the gene names have not been highlighted. We hope the revision addresses your concerns satisfactorily.

Looking forward to your responses,

Tin Tin Su & Monn Myat

Referee: 1

Myat and colleagues screen 30 RNAi lines for salivary gland migration defects and 15 lines for cell translocation defects after wing disc irradiation. The work is neither a screen nor an investigation of a developmental event or gene. With such a tiny selection of RNAi lines one cannot judge if the reported mild phenotypes are real or not (especially when 13 out of 15 show a phenotype as in the irradiation experiment).

*We apologize that the intent of the study was unclear. The point was not to perform an unbiased screen through all *Drosophila* genes, which, as the reviewer correctly point out, should give hit rates of <10%. The purpose was to ask, from among genes with known or suspected roles in cell movements, which are required for salivary gland morphogenesis and which are required during disc regeneration. This, we hypothesized, would allow us to compare the requirement for the same cell biological process in the same organism, but in two different contexts. Therefore, we started with a focused group of genes already known to be important for cell morphology, cytoskeleton function and cell movement but in other contexts. We make this clear in the revised Supplemental Table 1.*

A focused approach is similar to what was done to identify epigenetic regulators of salivary gland cell death (~100 RNAi lines, some redundant for the same target, against genes known to have epigenetic function, rather than an un-biased collection; PMID: 25211335); another to identify ER proteins involved in early secretory pathway (156 lines tested; PMID: 21383842); or focused screen through genes with similar biochemical functions (e.g. just the kinases).

We do appreciate the reviewer's concerns. To avoid confusion, we have removed 'screen' from the manuscript including the title.

In addition, there are technical issues, e.g. there is no quantification for the salivary gland screen data, and for the wing disc screen 2 wing discs from the same larva appear to have been used, further reducing the statistical significance (down to 4 independent data points).

Quantification of the salivary gland screen data is now provided in lines 125-129, lines 159-162, and in Figure 4D.

For the wing disc assay, the reviewer's assessment of independent data points is not correct and we apologize for our failure to provide a clear description of our methods. We dissect a minimum of 10 larvae per genotype and mount all discs onto a slide. We then randomly select and image 7 whole, flat discs for quantification (i.e. skipping those that ripped during dissection or became folded/crumpled during mounting). We choose a sample size of 7, we used a simplified resource equation from 'Charan J, Kantharia ND. How to calculate sample size in animal studies? J Pharmacol Pharmacother. 2013;4(4):303-6'. Briefly, E

= Total number of animals – Total number of groups, where E value of 10-20 is considered adequate. When we compare two groups (vector vs RNAi, for example), 6 per group or E = 11 would be adequate. Therefore, we chose 7. These explanations have been added to the Methods section (lines 426-431).

I would believe these hits only if a random selection of 30 or 15 RNAi lines results in 0-1 hits in the salivary gland and wing disc assays. 20-87% hits out of the total is not a screen.

The use of a selected set of genes with known function in cell movement, we believe, can explain why the hit rate is high. We agree that a random selection of 30 or 15 RNAi lines would give much lower hit rate, but that was not our study design. And as mentioned in a preceding response, we have removed 'screen' from the manuscript.

Minor comments:

I cannot understand how these 30 RNAi lines were chosen. A rationale should be provided (e.g. line 1 had phenotype x in screen y) in an extra column. Also, why many other hits from the referenced screens were not chosen.

The requested information has been added in extra columns in the revised Supplemental Table 1. Rationale for the lines used is now included in lines 119-125; all are insertions on chromosome II, which was preferred for technical reasons.

Line 125: Supplemental Table 1 should be referenced

Supplemental Table 1 was referred to in line 119 (line 122 in the revised text). Table S1 now has references cited within it.

Line 132: here and throughout the manuscript, gene names should be consistent and correspond to official FlyBase nomenclature, e.g DGK is rdgA, pdsw is ND-PDSW, enabled is ena etc.

We have harmonized the gene names.

Line 184: what about MAPK-AK2 apical domain defects?

MAPK-AK2 mutants were not analyzed for apical domain defects because there were no gross changes in lumen size or morphology observed by phalloidin staining for F-actin (Figure 1E). Only ena and ND-PDSW showed obvious lumen defects and were analyzed for apical domain defects. This information has been added to lines 194-195.

Line 388: ...is still oriented dorsally...

Corrected.

Line 441: are we talking about n=7 larvae or n=7 wing discs from 4 larvae? One would expect very similar results for 2 wing discs coming from the same larva.

Please see the full explanation in a preceding paragraph. Briefly, we mount on each slide, discs from at least 10 larvae (20 discs) and we image the first 7, so it is more likely to be n=7.

There is quite a bit of discussion in the last part of the results section.

We have now moved part of the Results into the Discussion and have rearranged the rest of the Results according to Reviewer 2's suggestion (see response below).

Supplemental table 1:

It would make more sense to have one column labeled symbol, and give in this column either the gene symbol or the CG number, then in column 2 the putative molecular function, e.g. diacyl glycerol kinase or actin, etc.

As it is right now, column 2 sometimes gives the gene name, sometimes the symbol, sometimes a mix.

We have re-organized this table, added extra information as explained in the preceding sections, and have harmonized the gene names.

Referee: 2

In this work, Myat et al. use different *Drosophila* models to investigate the contribution of selected genes with potential roles in collective cell migration. On the one hand they use the embryonic salivary gland, a well-established model, and on the other they analyze cell repositioning in wing disc regeneration. Their experiments shed light into the mechanism of this type of cell position change during regeneration. In addition they identify new genetic requirements for migration in salivary glands and in disc regeneration. A small part of the genes tested are shown to be required for migration in both systems. They conclude that different types of cell movements have common and specific genetic requirements.

The analysis presented allows the identification of new players in salivary gland formation and in regeneration. However, besides this descriptive (phenotypic) analysis, no molecular mechanisms are provided or presented.

Here I suggest different aspects that I consider that could help to improve the manuscript

1. Ena activity. The authors show that in *ena* mutants the distal part of the gland turns posteriorly but the proximal part does not. They observe lack of cell rearrangements in the proximal part. It is known that salivary gland distal cells extend actin-based protrusions. As the authors mention, it is also known that Ena can promote actin-based protrusions. These evidences raise the hypothesis that Ena promotes actin protrusions at the distal region and that this promotes proper migration/pulling force of the "leading" (distal) region that could indirectly promote cell rearrangements in the proximal part. Can the authors analyze actin protrusions in *Ena* mutants and compare to the wild type?

We did not see a difference in actin localization between ena mutants/RNAi and wild type based on phalloidin staining. These data are in Figure 1B-C. The caveat, however, is that phalloidin staining may not be sensitive enough to observe actin protrusions. Live imaging, for example, with GFP-actin crossed into the same genetic backgrounds used here may be required. Unfortunately, the senior author responsible this part of the work (Myat) has closed her lab and no longer has access to equipment capable of imaging actin protrusions in live embryos. We sincerely apologize that we cannot perform these additional experiments. We do concur with the reviewer that the main thrust of this work is to document the differential requirement of the same set of genes in two different cell movement systems and to identify any new players, leaving the mechanistic dissections to future studies.

2. *Ena*, MAPK-AK2, DGK and *pds* produce a similar salivary gland phenotype, however, at the cellular level (cell rearrangements) the effects seem quite different. Could the authors elaborate on whether they consider that these genes belong to 2 different mechanisms? Would MAPK-AK2, DGK and *pds* participate in a single mechanism?

Mechanisms by which these mutants could affect gland migration is now discussed in lines 310-344.

3. Ena localises apically and is enriched in some basal domains. In Ena mutants the authors show defects of lumen shape. They find that apical area is maintained but cells are not elongated along the proximo-distal axis as in the wild type. This effect is also observed in pdsw downregulation, although the mechanisms may be different. Do they also detect a similar effect downregulating MAPK-AK2, DGK? Are tubes, in general, shorter and wider?

Among ena, ND-PDSW, rdgA and MAPK-AK2, only the first two show obvious lumen defects. The lumens appeared normal in the last two. Therefore, we analyzed apical domain elongation only in the first two. This information has been added to lines 194-195.

4. The authors report defects in cell shape in Ena and in pdsw mutants. Could these defects underlie the phenotype of salivary glands observed, rather than being a consequence of impaired collective cell migration? How can the authors distinguish between defects in migration or defects in cell shape during salivary gland migration?

Discussion of the underlying cause of the gland defects in ena and pdsw mutants is now included in lines 346-354.

5. I find the results section about wing disc regeneration a bit confusing.

I suggest to divide the chapter into a first one where they show that cell repositioning relies on cell migration as well as directed divisions. This is a very interesting result by itself and helps to interpret the rest of results. Another chapter/s could document the effect of downregulation of the different genes tested in cell repositioning. I also feel that some aspects in this results section could be transferred to the discussion section.

We have rearranged the Results (and revised Figure 5) according to this recommendation. Per suggestion from this reviewer, we include a new section on the role of cell division starting in line 197. A new section on downregulation of the different genes now start in line 258. The effect of Rux is thus presented first in its own section, before the effect of RNAi lines. Some parts of the Results have also been transferred into Discussion. Thanks for the suggestion.

6. Surprisingly, from all lines tested in wing disc regeneration, all except 2 produce an effect. These lines are precisely two of the ones identified in salivary gland experiments. In addition, the rest of lines required for salivary gland formation produce mild effects in wing disc regeneration. Furthermore, the top hits in wing disc produce no effects in salivary glands. Wouldn't these results suggest that cell movements during salivary gland development and during wing disc regeneration after IR damage depend on rather distinct genetic requirements?

We agree with the reviewer's interpretation of the results in general and have now incorporated similar language into the Discussion (lines 296-301). Because three genes identified in the salivary gland experiments (DGK, Rac2 and MAPK-AK2) did show defects in the wing disc with p values in the E-04 range, we do believe there is some overlap in genetic requirements between the two experimental systems. Our conclusions (lines 306-308) reflect this.

Appendix B

Dear Editor,

We thank referees 1 and 2 for their assessment that the previously revised version of the manuscript addressed their concerns adequately. Below, we address the concerns of referee 3 in a point-by-point response. Changes are highlighted in the text.

Sincerely,

Tin Tin Su and Monn Myat

Referee: 3

My biggest concern is the authors' interpretation of RNAi phenotypes in embryonic salivary glands. It is not uncommon that knocking down essential genes zygotically in fly embryos does not provide an overt phenotype due to strong maternal contribution. The authors say that they observed 8 out of 30 RNAi lines show migration defects of salivary glands, but it does not imply that the rest of 22 genes do not have a role in salivary gland migration. It is possible that maternally provided gene products were enough to allow the salivary gland to migrate properly without any overt migration defects. Indeed, high throughput expression data in Flybase shows "high expression" of *Past1*, "very high expression" of *Ankle2* in 0-2 hr embryos. These are the two genes that they mention in Discussion as being the top hits in the wing disc but showing no effect in the salivary gland (line 300). It is also true for *Dia* (moderate expression), *Arp2* (high expression) and *Act5c* (extremely high expression), another three genes that they mention in line 328 as the same cases. I did not look up the expression level of every single gene that the tested in this study, but I suspect that many genes would be the same as most of them are essential genes that have important functions in various cell and developmental contexts. The authors' conclusion on this paper is based on their interpretation of which RNAi lines show migration defects in either system, and I can't agree with it. I strongly suggest that the authors should consider and provide an alternative interpretation on their RNAi data in salivary glands. It is even possible that almost every single gene might have a role in salivary gland migration as it does in their wing disc translocation assay.

They also only tested a single RNAi for each gene, which further raises a concern that one won't be able to make a conclusion whether the absence of the phenotype is because those genes were not required for salivary gland migration or because these RNAi lines did not successfully knock down those genes.

Ankle2 is maternally deposited at 111 RPKM (Reads Per Kilobase of transcript, per Million mapped reads) in 0-2 hr embryos and its levels decline to 45 RPKM in 8-10 hr (stage 12) embryos, the relevant stage for salivary gland formation (gene_rpkms_report_fb_2017_05.tsv downloaded from Flybase on 03/29/2019). The corresponding values in the same dataset for genes that show RNAi phenotypes in the salivary gland are: PDSW, 131/121; MAPK-AK2, 92/57; Rab35, 58/32; Rab40, 51/13; Cdk9 32/28; ena, 22/34; and Rac2, 12/36. The data for *rdgA*, 0/1 is from BDGP in situ homepage (<http://insitu.fruitfly.org/cgi-bin/ex/report.pl?ftype=1&ftext=FBgn0261549>). In other words, while *Ankle2* RNAi did not show a salivary gland phenotype, PDSW and MAPK-AK2 that have comparable or greater transcript levels did. The finding that ANKLE2 RNAi showed a phenotype in wing discs means also that RNAi was effective. Therefore, RNA abundance or RNAi efficiency on their own cannot explain why RNAi against some genes produced a phenotype in the wing disc but not in salivary glands. RPKM values in 0-2 hr and 8-10 hr embryos for genes that showed RNAi phenotypes in the wing disc but not in the salivary gland are; *past1*, 65/53; CG2794, 6/13; *Arp2* 88/45; *dia*, 15/12; and *Act5C*, 1400/1500. These include genes such as *dia* and CG2794 with lower RPKM counts than the ones that did show RNAi phenotypes in the salivary gland, further supporting the idea that RNA abundance alone cannot explain why RNAi against some genes produced an effect in the wing disc but not in salivary glands.

The point raised by the reviewer is a good one, so we have included a shortened version of the above paragraph in the Discussion (lines 305-315).

Minor comments:

1. The authors showed mild salivary gland migration defects with 8 genes. This is a quite subtle phenotype and should be described/quantified better.

a. Figure 1. It took me a little while to clearly understand what the posterior migration defects are. The figure legends say arrow and arrowheads in there, but they are not shown in the images.

We lost the layer with arrows and arrowheads during revision. It has been restored.

b. How many salivary glands were analyzed for this phenotype? And what is the penetrance of the phenotype? Both

the number of glands and the penetrance of the phenotype should be included. They mentioned that they looked at live embryos first and then confirmed it with fixed samples when they observe more than 5 live samples with abnormal morphology. But it is not clear how many stage 14 salivary glands they quantified for the proximal migration defects.

In both live and fixed embryos, approximately 20 stage 14 glands each were analyzed. The penetrance in fixed samples was such that we detected 5-10 defective glands per sample. This information has been added to the revised text (line 129-130).

c. It is my understanding that *Drosophila* embryonic salivary glands migrate actively through stages 13-16. Do these proximal migration defects persist at later stages as well? Or is this something transient that is only observed at stage 14? If the latter, could it be a slight delay of migration?

We have not followed the glands beyond stage 14.

d. Lines 159-162: The cell rearrangement defects are only described in the text but never shown in the figure. It could help readers understand the defects better if images are provided as additional panels in the figure. I understand that the senior author who is responsible for this part does not have access to the equipment anymore, but it could be done easily if the original confocal data are still available.

The counting of nuclei around the lumen has been described in a publication from our lab; it is now cited as ref 20. Nuclear density changes for an allele of *ena* is shown in Supplemental Figure 1.

2. *Ena* loss of function phenotype

a. Line 186: No effects on localization of DaPKC and DE-Cad in *ena* mutants also could be due to maternal contribution.

It is formally possible, but because we do see defects in *ena* mutants (e.g. Figure 1C), we think this explanation is unlikely.

b. Figure 4. It should be better described how the apical area was measured. They cited previous papers where the same quantification had been done in the Method section, but adding arrows along the A-P and D-V axis in the image (or a cartoon) to show what is measured would help.

We have added a cartoon to revised Figure 4D.

3. Directed cell divisions and wing shape

a. Line 236: "randomizing of division orientation produces a round wing (29)" It is my understanding that the reference 29 showed a correlation between oriented cell division and the wing shape, but did not show the causal relationship. On the contrary, a very recent paper [Zhou et al., (2019) Oriented Cell Divisions Are Not Required for *Drosophila* Wing Shape. *Curr Biol* 29: 856-864] showed that oriented cell divisions are not required for *Drosophila* wing shape.

Actually, Ref29 includes data on the causal relationship in that *ds* mutants show random orientation of mitotic spindles in wing discs and altered wing shape from oval to round. The new paper used a different mutant, in *Drosophila* NuMA homolog *mud*, to show that even when spindle orientation was randomized, normal wing shape can result due to a compensatory mechanism that involves cell arrangements. These data may be reconciled as spindle orientation matters but compensatory mechanisms also contribute to wing shape. The text has been modified to reflect this point in lines 239-240. Thanks for alerting us of the 2019 paper, which is now cited as ref 31.

b. Lines 295-296: "Of 30 RNAi lines tested in the salivary glands, eight produced defects and included two known and four novel regulators." If two are known regulators and four are novel, what are the remaining two out of eight?

Corrected to three known regulators (*Rac2* and two *Rab* proteins) and four novel regulators (lines 298-300). The remaining one is *Cdk9*, which is likely to be an indirect regulator with pleiotropic effects.